# Target spike patterns enable efficient and biologically plausible learning for complex temporal tasks

Paolo Muratore[1]*, Cristiano Capone[2], Pier Stanislao Paolucci[2]

1 SISSA—International School for Advanced Studies, Trieste, Italy, 2 INFN, Sezione di Roma, Rome, Italy

☯ These authors contributed equally to this work.
* pmurator@sissa.it

**Data Availability Statement:** The python code for reproducing the obtain results is published in the GitHub public repository (https://github.com/myscience/LTTS).

## Abstract

Recurrent spiking neural networks (RSNN) in the brain learn to perform a wide range of perceptual, cognitive and motor tasks very efficiently in terms of energy consumption and their training requires very few examples. This motivates the search for biologically inspired learning rules for RSNNs, aiming to improve our understanding of brain computation and the efficiency of artificial intelligence. Several spiking models and learning rules have been proposed, but it remains a challenge to design RSNNs whose learning relies on biologically plausible mechanisms and are capable of solving complex temporal tasks. In this paper, we derive a learning rule, local to the synapse, from a simple mathematical principle, the maximization of the likelihood for the network to solve a specific task. We propose a novel target-based learning scheme in which the learning rule derived from likelihood maximization is used to mimic a specific spatio-temporal spike pattern that encodes the solution to complex temporal tasks. This method makes the learning extremely rapid and precise, outperforming state of the art algorithms for RSNNs. While error-based approaches, (e.g. e-prop) trial after trial optimize the internal sequence of spikes in order to progressively minimize the MSE we assume that a signal randomly projected from an external origin (e.g. from other brain areas) directly defines the target sequence. This facilitates the learning procedure since the network is trained from the beginning to reproduce the desired internal sequence. We propose two versions of our learning rule: spike-dependent and voltage-dependent. We find that the latter provides remarkable benefits in terms of learning speed and robustness to noise. We demonstrate the capacity of our model to tackle several problems like learning multidimensional trajectories and solving the classical temporal XOR benchmark. Finally, we show that an online approximation of the gradient ascent, in addition to guaranteeing complete locality in time and space, allows learning after very few presentations of the target output. Our model can be applied to different types of biological neurons. The analytically derived plasticity learning rule is specific to each neuron model and can produce a theoretical prediction for experimental validation.

**Funding:** This work has been supported by the European Union Horizon 2020 Research and Innovation program under the FET Flagship Human Brain Project (grant agreement SGA3 n. 945539 and grant agreement SGA2 n. 785907) and by the INFN APE Parallel/Distributed Computing laboratory. Fundings were received by P.S. Paolucci. The funders had no role in study design, data collection and analysis, decision to publish, or preparation of the manuscript.

**Competing interests:** The authors have declared that no competing interests exist.

## Introduction

The development of biologically inspired and plausible neural networks has a twofold interest. On the contrary, neuroscience aims to achieve a better understanding of the functioning of biological intelligence. On the other hand, machine learning (e.g. deep learning [1]) tries to borrow secrets from biological networks. To justify the search for biological learning principles it is enough to consider that the human brain works with a baseline power consumption estimated at about 13 watts, of which 75% spent on spike generation and transmission [2].

The transmission of information through spikes is a widespread feature in biological networks and is believed to be a key element for efficiency in energy consumption and for the detection of causal relationship between events. Spike-timing-based neural codes are experimentally suggested to be important in several brain systems. In the barn owl auditory system, for example, coincidence-detecting neurons receive temporally precise spike signals from both ears [3]. In humans, precise timing of first spikes in tactile afferents encodes touch signals at the fingertips [4]. If the same touch stimulus is repeated several times, the relative timing of action potentials is reliably reproduced [4]. Similar coding have also been suggested in the rat's whisker response [5] and for rapid visual processing [6].

Because of the biological relevance of spike-time-based coding, in the last years different spiking network models have been proposed, for feedforward networks [7–11] and for recurrent ones [12–16].

An important step towards biological plausibility has been to show that a backpropagation-like algorithm works even if the feedback matrix is random and fixed [9]. A similar principle can be used to obtain a biologically plausible approximation of BPTT and train recurrent neural networks [12, 17].

However, the capture of long-time temporal dependence on backpropagation-based techniques is computationally expensive and leads to synaptic updating rules that are difficult to frame in a biological substrate. Target-based learning frameworks [14, 15], as opposed to error-based [12, 18], bypass the issue of biologically implausible error back-propagation by providing to the network a target activity to mimic.

A first approach to implement a target based training consists in inducing in the recurrent network, through an external stimulus, a spatio-temporal pattern of activity, which is learned by the network through Hebbian plasticity [16, 19, 20]. Other approaches propose to leverage the intrinsic dynamic of the network by imposing one chaotic sequence produced by the network as an internal target. This strategy has been explored for both rate [21] and spiking neurons [22]. Interestingly, target-based approaches have also been proposed for feedforward networks, where targets, which propagate instead of errors, can be assigned to each layer of the deep network [23].

In this work, we provide a theoretical framework based on maximum likelihood techniques [24, 25] for target-based learning in biological recurrent networks. We show that the introduction of a likelihood measure as objective function is sufficient to derive biologically realistic target-based synaptic updating rules.

Other papers have already proposed probability-based approaches assuming that learning in the recurrent network consists of adapting all the synaptic weights such that the Kullback-Leibler divergence between the target activity and the generated one is as small as possible (or similarly such that likelihood of the target activity is as large as possible) [26–29].

However, these probability-based approaches (maximum likelihood or minimum Kullback-Leibler divergence) were mainly aimed to train the network to reproduce specific patterns of spikes, while we show how to use randomly projected target patterns of spikes to solve complex temporal tasks.

We propose a target-based learning protocol that assigns to the RSNN a specific internal spike coding for the solution of a task, instead of looking for a solution among those in a huge space of possible alternatives, as error-based approaches would do.

The choice of a target-based learning protocol was already described in [14, 15] as the full-FORCE algorithm. The major difference of our model is that we propose as a target a specific spatio-temporal spike pattern, allowing for precise spike timing coding. For this reason we named our learning protocol LTTS (Learning Through Target Spikes). This particularity of LTTS allowed us to achieve tremendous efficiency in learning and precision in the performance of the tasks. In addition, our approach turns out to be very natural in terms of biological plausibility. Indeed it does not require the assumption of error propagation in the cortex, which is not yet a solid feature in terms of experimental observations. Rather, we rely on the presence of the so termed 'referent activity templates' which are spike patterns generated by neural circuits present in other portions of the brain, which are to be mimicked by the network subjected to learning [30, 31]. Also, the learning rule emerging from our likelihood-based framework results to be biologically plausible because it is local to the synapse, namely it does not require the evaluation of global observables over multiple neurons or multiple times.

Furthermore, in order to define a learning rule which is completely local in space and time we perform an online approximation of the gradient ascent [28], which demonstrates to be extremely beneficial to the training velocity when learning from a small number of presentations.

We use our learning protocol to train a RNN to solve 3 different tasks. First, we train the network to reproduce 3D trajectories. We achieved an error which is lower than the ones achieved by e-prop1 [12] and by the biologically unplausible BPTT. Second, we trained the network to reproduce a high dimensional (56 dimensions) walking dynamics trajectory [32], showing that it is able both to learn the sequence and to display a spontaneous walking pattern beyond the learned epoch. Third, we demonstrate that our model is able to solve the classical temporal XOR task.

## Results

### Target-based training protocol

We considered a target-based approach which conceptualizes learning as the successful replay of a target sequence of spikes $s_{\text{targ}} = \{s_{i,\text{targ}}^t\}$ where $s_{i,\text{targ}}^t$ is 1 whether the neuron $i$ emitted a spike at time $t$ and 0 otherwise. A very important point is the way such $\{s_{i,\text{targ}}^t\}$ is defined. Compared to unsupervised or reward-based learning paradigms, supervised paradigms on the level of single spikes might appear less relevant from a biological point, since it is questionable what type of signal could tell the neuron about the target spiking sequence. Here we propose a biologically plausible way to generate such target. The neurons in the RSNN to be trained receive a randomly projected input $\boldsymbol{I}_{\text{teach}}^t$ from neurons belonging to another population of neurons, e.g. from other cortical areas [16, 33]. $\boldsymbol{I}_{\text{teach}}^t$ shapes the probability of firing, defining the target spiking sequence $s_{i,\text{targ}}^t$.

### The target pattern of spikes

The idea is to construct an internal target pattern of activity $\{s_{i,\text{targ}}^t\}$ for the spiking network that contains the relevant information needed to solve the task, namely a pattern that encodes the target output signal.

Let's now suppose that the teaching signal is a function (e.g. a linear function $\boldsymbol{I}_{\text{teach}}^t = \boldsymbol{J}_{\text{teach}}\boldsymbol{y}^{\text{in}}$) of the activity $\boldsymbol{y}^{\text{in}}$ of other areas. The objective of our protocol is to train the network to produce

a target readout $y^{\text{targ}} = J_{\text{out}} s_{\text{targ}}$, in principle accessible to another area of the brain. In conclusion the training phase is obtained under the influence of $y^{\text{in}}$ coming from some areas (e.g. a visual sequence in the visual cortices) and influencing the internal target $s_{\text{targ}}$, while in the retrieval the network is autonomously able to provide the output signal $y^{\text{targ}}$ (e.g. a motor sequence in the motor cortices). E.g. the target might be to learn a motor task ($y^{\text{targ}}$) through visual observations ($y^{\text{in}}$). It is a quite natural hypothesis that the activity $y^{\text{in}}$ composing the teaching signal is a function of target output $y^{\text{in}} = F(y^{\text{targ}})$ (and vice versa). In this work we consider the simplest scenario where $y^{\text{in}} = y^{\text{targ}}$, as represented in Fig 1A where $J_{\text{teach}}$ projects the $y^{\text{targ}}$ signal.

This choice makes the internal target correlated to the target output $y^{\text{targ}}$ making easy the training of the readout weights $J_{\text{out}}$.

We define the internal target $s_{i,\text{targ}}^t$ as the pattern of spikes induced in the network by the training current $I_{\text{teach}}^t$, in absence of recurrent connections, possibly in combination with an additional input signal that, depending on the task, can be either superfluous or essential. In the memory recall tasks the latter may serve as an external clock, in others (e.g. XOR task) it is the input to be processed to solve the task.

The importance for the choice of a specific internal coding $\{s_{i,\text{targ}}^t\}$ can be clarified after considering the alternative error-based approach [12]. In that case, the space of the solution is extremely high, namely a large number of different internal coding are capable to solve the task. On the other hand, the choice for the specific internal solution $\{s_{i,\text{targ}}^t\}$ makes the learning extremely faster.

In terms of biological plausibility, we can assume that the target pattern of spikes is computed by a dedicated compartment of the neuron that receives the training current. This would be totally equivalent to our protocol. The target activity can be locally computed at every trial of the training procedure, and it is always accessible to the network, without the requirement of long-term storage. We consider this assumption biologically plausible because of the increasing consensus on the specific computational role of the apical compartments of pyramidal neurons [33] which receive contextual signals from other areas.

Such formalism can be applied to accomplish different temporal tasks. In the next sections, we first introduce the learning algorithm, then we apply it to several complex tasks.

## Network model

We propose a network of spiking neurons described by the real-valued variable $v_j^t \in \mathbb{R}$, their membrane potential, where the $j \in \{1, \ldots, N\}$ label identifies the neuron and $t \in \{1, \ldots, T\}$ is a discrete time index. Each neuron exposes an observable state $s_j^t \in \{0, 1\}$, which represents the occurrence of a spike from neuron $j$ at time $t$ and it is randomly generated as a function of the membrane potential $p(s_j^{t+1}|v_j^t)$. We use $v^t = \{v_j^t\}$ and $s^t = \{s_j^t\}$ to indicate the collections of internal and observable network states.

Assuming a synchronous update dynamics, the likelihood of a target sequence of spikes $\{s_{\text{targ}}^t\}$ can be easily written as the product over all times and neurons of the probability $p(s_{j,\text{targ}}^{t+1}|v_j^t)$. Thus the log-likelihood of the network activity $s_{\text{targ}}$ can be introduced as:

$$
\begin{aligned}
\mathcal{L}(s_{\text{targ}}; J) = \log p(s_{\text{targ}}; J) &= \log \prod_{t=1}^{T}\prod_{i=1}^{N} p(s_{\text{i,targ}}^{t+1}|v^t; J) \\
&= \sum_{t=1}^{T}\sum_{i=1}^{N} \log p(s_{\text{i,targ}}^{t+1}|v^t; J)
\end{aligned}
\tag{1}
$$

where we made explicit the dependence on $J$, the collection of parameters defining the model.

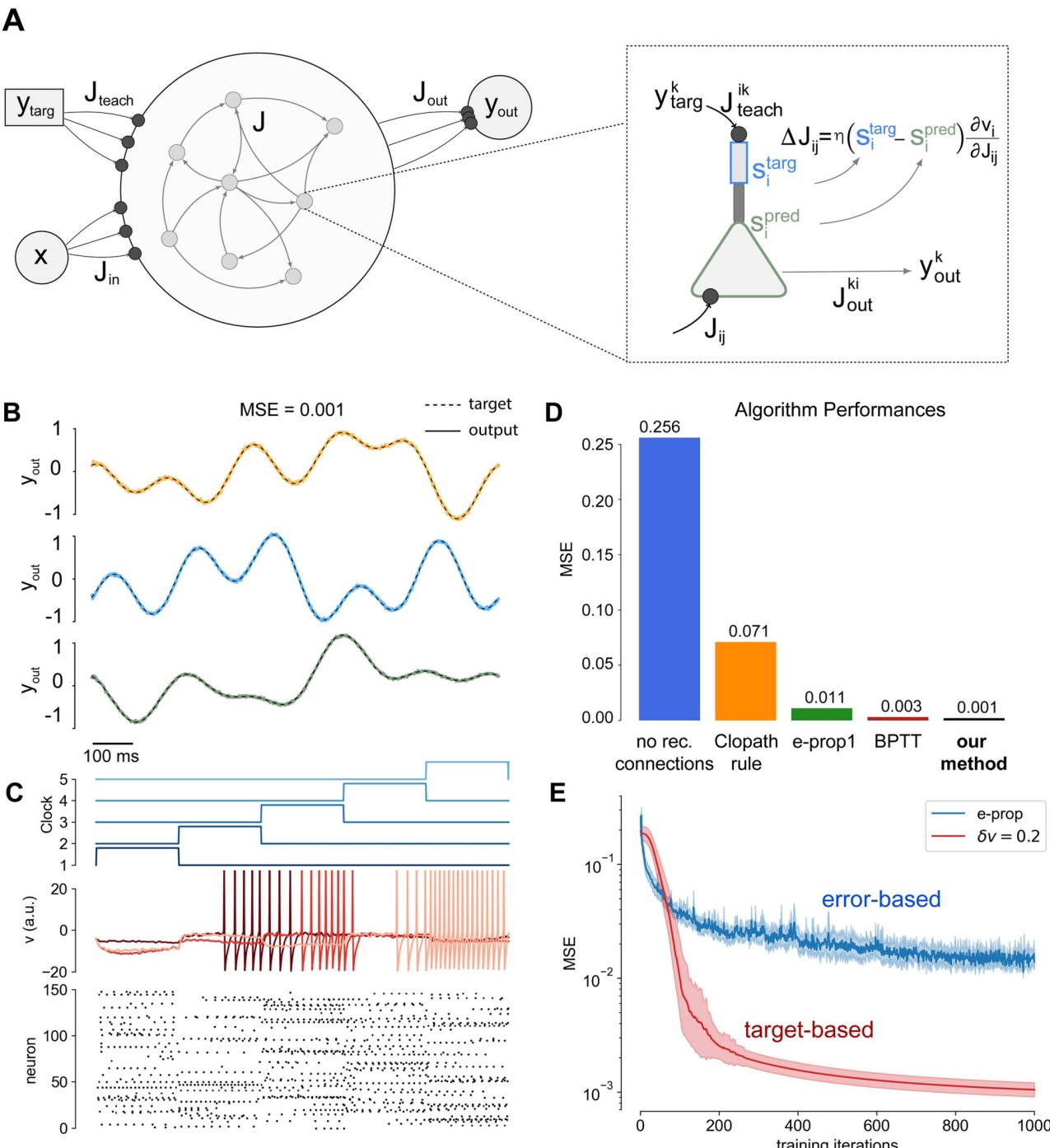

**Fig 1. Results for the Pattern Generation task. A)** Schematics of the system's architecture. Each neuron is composed of two compartments where the basal region (green) elaborates a prediction, while the apical part (blue) receives the teaching input and computes the target: the learning rule depends on the difference between this two quantities. The recurrent activity is decoded via a linear readout layer with three output neurons, one for each trajectory. **B)** Dashed lines: the target output signal. Solid lines: output signal generated by the network. An average MSE (Mean Squared Error) of MSE = 0.001 was achieved. **C)** (top) The clock input: it ranges between zero and one (vertical shift for visual purposes). (middle) plot the dynamics of the internal state of six neurons randomly extracted from the network population is reported over time. (bottom) The raster plot of the activity of a population of 60 neurons from the complete network. The simulation used a network of $N = 500$ neurons and a total simulation time of $T = 1000$ time steps. **D)** Comparison of different learning algorithms on the Pattern Generation Task. Learning performances are evaluated as final MSEs. The blue bar represents the final MSE when only the readout layer is trained, thus no recurrent synapses optimization is performed. Performances of the Clopath [35] learning rule, e-prop1 [12] and Back-Propagation-Through-Time (BPTT), are reported respectively in orange, green an brown. **E)** MSE as a function of the number of training iterations, comparison between our model (red) and e-prop (blue).

The probability of spike generation can be written as:

$$p\left(s_i^{t+1}|v_i^t\right) = \frac{\exp\left[s_i^{t+1}\left(\frac{v_i^t-v^{th}}{\delta v}\right)\right]}{1 + \exp\left(\frac{v_i^t-v^{th}}{\delta v}\right)} \tag{2}$$

where $v^{th}$ is the firing threshold and $\delta v$ defines the amount of noise in the spike generation. In this work, we considered the effect of different $\delta v$ values on plasticity and their effects on the learning rate and on the external perturbations. The spike generation is always performed in the deterministic limit $\delta v \to 0$ in which Eq (2) becomes $p(s_i^{t+1}|v_i^t) = \Theta[s_i^{t+1}(v_i^t - v_i^{th})]$. This approximation on the spike generation does not impair the convergence of the algorithm and gives remarkable benefits in terms of learning speed and robustness to noise.

In the left-hand side of Eq (1) the dependence on $\boldsymbol{v}^t$ is not explicit. This is because $\boldsymbol{v}^t$ depends on $\boldsymbol{s}^t$, and is uniquely defined by setting a specific initial condition $\boldsymbol{v}^t = \boldsymbol{v}^0$ and the following rule:

$$\boldsymbol{v}^t = D(\boldsymbol{v}^{t-1}, \boldsymbol{s}_{\text{targ}}^t) \tag{3}$$

where $D(\boldsymbol{v}^{t-1}, \boldsymbol{s}_{\text{targ}}^t)$ is a generic function and depends on the chosen model of neuron.

## Learning rule for cuBa neurons

Without loss of generality (see the section dedicated to conductance based neurons **Learning rule for coBa neurons**) we now assume for simplicity that the neuron integrates its input as a current-based (cuBa, the integration of synaptic input does not depend on the membrane potential itself) leaky integrate and fire (LIF) neuron. The membrane potential of the neuron follows the equation:

$$\boldsymbol{v}^t = \left(1 - \frac{\Delta t}{\tau_m}\right)\boldsymbol{v}^{t-1} + \frac{\Delta t}{\tau_m}(\boldsymbol{J}\hat{\boldsymbol{s}}^t + \boldsymbol{I}^t + v_{\text{rest}}) - J_{\text{res}}\boldsymbol{s}^t \tag{4}$$

where $\Delta t$ is the time integration step, $\tau_m$ is the membrane time constant, $\boldsymbol{J} \in \mathbb{R}^{N \times N}$ is the recurrent network matrix that defines the synaptic connections within our model, while $\boldsymbol{I}^t$ is a general external input. $v_{rest}$ is the rest potential, the asymptotic value of the membrane potential in absence of input currents. $J_{res} = 20mV$ accounts for the reset of the membrane potential.

We also assumed that the spikes undergo an exponential synaptic filtering with a time scale $\tau_s$. $\hat{\boldsymbol{s}}^t$ is the filtered spike signal and is expressed as:

$$\hat{\boldsymbol{s}}^t = \left(1 - \frac{\Delta t}{\tau_s}\right)\hat{\boldsymbol{s}}^{t-1} + \frac{\Delta t}{\tau_s}\boldsymbol{s}^t \tag{5}$$

The derivation can be generalized to different choices of synaptic filtering kernels (e.g. alpha function). To make the recurrent network reliably reproduce the target pattern of activity we need to maximize the likelihood $\mathcal{L}(\boldsymbol{s}_{\text{targ}}; \boldsymbol{J})$ for the system to express the dynamics defined by $\boldsymbol{s}_{\text{targ}} = \{s_{i,\text{targ}}^t\}$.

This can be achieved by a gradient based algorithm, which suggests that the optimal plasticity rule is a weight update proportional to the likelihood gradient (see Methods section for derivation):

$$\Delta J_{ij} = \eta \frac{\partial \mathcal{L}(\boldsymbol{s}_{\text{targ}}; \boldsymbol{J})}{\partial J_{ij}} = \eta_0 \sum_{t=1}^{T-1}\left[s_{i,\text{targ}}^{t+1} - s_{i,\text{pred}}^{t+1}\right]\frac{\partial v_i^t}{\partial J_{ij}} \tag{6}$$

were $\eta_0 = \frac{\eta}{\delta v}$. $\eta$ is scaled proportionally to $\delta v$ to keep $\eta_0$ constant. $s_{i,\mathrm{pred}}^t$ is the activity predicted by the network without the teaching signal. This stands in the $\delta v \to 0$ limit (the case for finite $\delta v$ is presented below, see Eq (7)). Indeed, in this limit Eq (2) becomes $p(s_i^{t+1}|v_i^t) = \Theta[s_i^{t+1}(v_i^t - v_i^{th})]$ and spike generation is deterministic. We notice that all the numerical experiments are performed accordingly to this definition of deterministic spike generation.

It is worth underlining the peculiar form of the obtained expression (Eq 6), which is remarkably similar to the form obtained in [34], where the first term in the left parenthesis can be regarded as an instantaneous prediction error: at each time step the target and spontaneous activity are compared and a learning signal is produced.

We assumed that each neuron in the recurrent network is composed of two compartments (see Fig 1A). The the apical one (blue) receives the teaching signal and computes the target $s_{i,\mathrm{targ}}^t$ while the basal one (green) elaborates a prediction $s_{i,\mathrm{pred}}^t$. The learning rule adjusts the recurrent weights to make the prediction coherent with the target.

The second factor $\frac{\partial v_i^t}{\partial J_{ij}}$ represents what is referred to in the literature as the spike response function [34]: it uses the information of the target trajectory to enable learning only for synapses which are causally related to recent pre-synaptic activity.

When the $\delta v \to 0$ limit is not taken we get the following voltage-dependent learning rule

$$\Delta J_{ij} = \eta_0 \sum_{t=1}^{T-1} \left[ s_{i,\mathrm{targ}}^{t+1} - f(v_i^t) \right] \frac{\partial v_i^t}{\partial J_{ij}} \tag{7}$$

where we defined $f(v^{\mathrm{th}}) = \frac{\exp \frac{v^t - v^{\mathrm{th}}}{\delta v}}{1 + \exp \frac{v^t - v^{\mathrm{th}}}{\delta v}}$. This version of the learning rule (where the deterministic limit is not taken), together with a deterministic generation of the spikes (the spike generation is always deterministic in all experiments described in this paper), provided a remarkable robustness to noisy perturbations (see **Robustness to noise** section below).

It is interesting to consider that while the former spike-dependent rule resembles the standard STDP synaptic update (as discussed in [26]), the latter voltage-dependent rule is coherent with what has been proposed in [35], where the plasticity depends on the membrane potential of the postsynaptic neuron.

Finally, in order to further improve biological plausibility, we used an approximation of the gradient ascent in which the weights are updated at each time bin $t$ accordingly to the following equation (for the spike-dependent case):

$$\Delta J_{ik}^t = \eta_0 \left[ s_{i,\mathrm{targ}}^{t+1} - s_{i,\mathrm{pred}}^{t+1} \right] \frac{\partial v_i^t}{\partial J_{ik}} \tag{8}$$

instead of updating them after the end of the training trial as expressed in Eq 6. The comparison between the gradient ascent and the online approximation is reported in the section **Learning with very few presentations**

## Learning 3D trajectories

To test the performances of our learning protocol we first tackle the standard three-dimensional temporal pattern generation task.

The task definition is the following: given an input signal $I_{\mathrm{clock}}^t$ (which we refer to as clock), the network is asked to produce an output target signal through a linear readout of the internal spiking activity. Namely the system is asked to learn a temporal input-output relationship between the clock signal and the target trajectory.

However, we remark that for this specific task the clock signal is not necessary (see S4 Fig in S1 File). A target sequence can be learned and recalled also without a temporally structured input (what we call clock). The target sequence is reproduced only by recurrent connections. In this case the sequence is initiated by providing the first spikes of the sequence. Our choice to use the clock is taken in order to reproduce exactly the same conditions proposed in the benchmark reported in [12, 13].

The target output $y^t_{\text{targ}} \in \mathbb{R}^3$ is a temporal pattern composed of 3 independent continuous signals (see Fig 1B, solid lines.). Each target signal is specified as the superposition of the four frequencies $f \in \{1, 2, 3, 5\}$ Hz with uniformly extracted random amplitude $A \in [0.5, 2.0]$ and phase $\phi \in [0, 2\pi]$. The network is also supposed to receive a clock-like input signal $I^t_{\text{clock}} = J^{\text{in}} x^t_{\text{clock}}$, where $J^{\text{in}} \in \mathbb{R}^{N \times K}$ and $x^t_{\text{clock}} \in \mathbb{R}^K$. The temporal structure of the clock is reported in Fig 1C(top).

For this task we equip our recurrent network with a standard linear read-out layer $y^t = J^{\text{out}} s^t$, where $J^{\text{out}} \in \mathbb{R}^{3 \times N}$ and $s^t \in \{0, 1\}^N$.

To produce a valuable target network activity we record its spontaneous activity, in absence of recurrent connections, when subject to an input $I^t = I^t_{\text{clock}} + I^t_{\text{teach}}$ composed of the clock-like term $I^t_{\text{clock}} = J^{\text{in}}_{\text{clock}} x^t$ and a random projection of the target signal $I^t_{\text{teach}} = J^{\text{in}}_{\text{teach}} y^t_{\text{targ}}$. In this scenario both $J^{\text{in}}_{\text{clock}}$ and $J^{\text{in}}_{\text{teach}}$ are static random Gaussian matrix with zero mean and variance $\sigma_{\text{in}}$ and $\sigma_{\text{teach}}$. Our choice was to set the teach $I^t_{\text{teach}}$ and the clock current $I^t_{\text{clock}}$ at comparable order of magnitudes (see Table 1 for the number of neurons and the other parameters used for this task).

We observe that the matrix $J^{\text{in}}_{\text{teach}}$ contributes to define the internal target pattern $\{s^t_{i,\text{targ}}\}$ since it induces such internal dynamics. The network is asked to learn to autonomously reproduce the target internal dynamics $\{s^t_{i,\text{targ}}\}$ by using the recurrent synapses $J$ and the clock signal $I^t_{\text{clock}}$ in absence of the teaching signal $I^t_{\text{teach}}$. This is achieved through the likelihood maximization, and the weight updates expressed in the previous section. Also the readout weight are trained with a standard minimization of the error function (see Methods section for details).

The readout layer is simultaneously trained to decode such information from the target pattern using standard optimization techniques (MSE objective function with Adam optimizer

**Table 1. Collection of model parameters used in the various tasks presented in the main text.** We have indicated with GA the gradient ascent algorithm.

|  | 3D Trajectories | Temporal XOR | Walking Dynamics | Few Presentations |
|---|---|---|---|---|
| $N$ | 500 | 500 | 500 | 500 |
| $T$ | 1000 | 130 | 600 | 50 |
| $\Delta t$ | 1 ms | 1 ms | 1 ms | 1 ms |
| $v^0_i$ | $-0.5 \; \forall i$ | $-0.5 \; \forall i$ | $-0.5 \; \forall i$ | $-0.5 \; \forall i$ |
| $s^0_i$ | $s^0_{i,\text{teach}} \quad \forall i$ | $s^0_{i,\text{teach}} \quad \forall i$ | $s^0_{i,\text{teach}} \quad \forall i$ | $s^0_{i,\text{teach}} \quad \forall i$ |
| $\eta/\delta v$ | 0.5 (Adam) | 0.5 (Adam) | 0.5 (Adam) | 1.0 (GA vs online) |
| $\delta v$ | $\rightarrow 0$ | $\rightarrow 0$ | $\rightarrow 0$ | $\rightarrow 0$ |
| $v^{th}$ | 0 | 0 | 0 | 0 |
| $\tau_s$ | 2 ms | 2 ms | 2 ms | 1.25 ms |
| $\tau_m$ | 8 ms | 8 ms | 8 ms | 2 ms |
| $\tau_{rout}$ | 20 ms | 2 ms | 20 ms | 20 ms |
| $\sigma_{teach}$ | 10 | 5 | 10 | 10 |
| $\sigma_{in}$ | 2 | 3 | 2 | 2 |
| $v_{rest}$ | $-4$ | $-4$ | $-4$ | $-1$ |
| $J^0_{rec}$ | $0 \; \forall i, j$ | $0 \; \forall i, j$ | $0 \; \forall i, j$ | $0 \; \forall i, j$ |

[36]), while the recurrent network exploits the maximum-likelihood training procedure to adjust its synaptic matrix $\boldsymbol{J}$ to generate a recurrent signal that, together with the clock-like input $\boldsymbol{I}_{\text{clock}}^{t}$, results in the desired temporal pattern.

In the retrieval phase the plasticity is turned off and the clock current is provided to the network. The spiking generated activity is decoded by the linear readout, and such output is compared to the target trajectory by evaluating the mean square error (MSE).

The system is very efficient to solve the task, interestingly after only 100 steps of the online Adam optimizer (similarly to the online gradient ascent the update is performed at each time step and not at the end of the trial) the system was able to reproduce the trajectory with an error Mean Squared Error MSE = 0.02 (we considered $\delta v$ = 0.2). In Fig 1B (dashed lines) we reported the results for the Pattern Generation task. After 1000 iterations of the online Adam optimizer the system achieved a final Mean Squared Error MSE = 0.0010±0.0003 (statistics computed over 50 realizations of different output targets, $\delta v$ = 0.2), computed across the three-dimensional output (for comparison $MSE_{e-prop}$ = 0.014±0.005). The activity of the internal neuron state $\boldsymbol{v}^{t}$, is reported over time, together with a raster plot of the recurrent network activity (Fig 1C). It is relevant to stress that the obtained performances are significantly better with respect to competing alternative algorithms for spiking neural networks learning temporal sequences. A quantitative comparison of performances, measured as final MSEs, is reported in the bar plot of Fig 1C. The blue bar represents the final MSE when only the readout layer is trained, thus no recurrent synapses optimization is performed. Performances of the Clopath [35] learning rule, e-prop1 [12] and Back-Propagation-Through-Time (BPTT), are reported respectively in orange, green an brown.

We also report the MSE achieved by our target-based protocol as a function of the number of training iterations (see Fig 1E, red line) average over 50 realization of the experiment) and compare it with the error-based e-prop (Fig 1E, blue line).

## Robustness to noise

A major consideration we addressed is the behaviour of our system in the presence of noise: how resilient is the learned dynamics under external perturbation?

To answer this question we considered both the spike-dependent learning rule (for $\delta v \rightarrow 0$, see Eq (6)) and the voltage-dependent one ($\delta v$ = 0.05, see Eq (7)).

The network is trained to generate the 3D trajectory (as discussed above). The resulting MSE as a function of the number of training iterations is shown in Fig 2 (blue lines). Then, we gauge the robustness of our trained model corrupting the input the network receives by injecting noise on top of the clock current as follows $\boldsymbol{I}_{\text{clock}}^{t} = \boldsymbol{J}_{\text{clock}}^{\text{in}}(\boldsymbol{x}^{t} + \sigma_{\text{noise}}\boldsymbol{\xi}(t))$, where $\boldsymbol{\xi}(t)$ is extracted randomly from a Gaussian with zero mean and unit variance. Statistics is evaluated over 625 realizations for each $\sigma_{\text{noise}}$ value. (25 different 3D trajectory times 25 realizations of the noisy input).

We measured the MSE between the target output and the trajectory generated by the perturbed network and the number of spikes not following the target sequence $\Delta S = \frac{1}{N\,T}\sum_{it}|s_{i,\text{targ}}^{t+1} - s_{i,\text{pred}}^{t+1}|$.

We observe a superior robustness for the voltage-dependent rule (see Fig 2B, blue line), with a much weaker robustness to noise for the spike-dependent one (see Fig 2A, blue line).

Also, the $\delta v$ parameter modulates the network tolerance to failed spikes or altered spikes timing: red lines in Fig 2A and 2B illustrate such behaviour. When $\delta v \rightarrow 0$ the output MSE is strongly tied to correct target retrieval: few noise-induced incorrect spikes are sufficient to immediately degrade performances and the task is failed (example $\sigma_{\text{noise}}^{2}/\sigma_{\text{clock}}^{2} > 0.01$). When $\delta v$ = 0.05 the system expresses strong resilience to spikes failing: under moderate noise

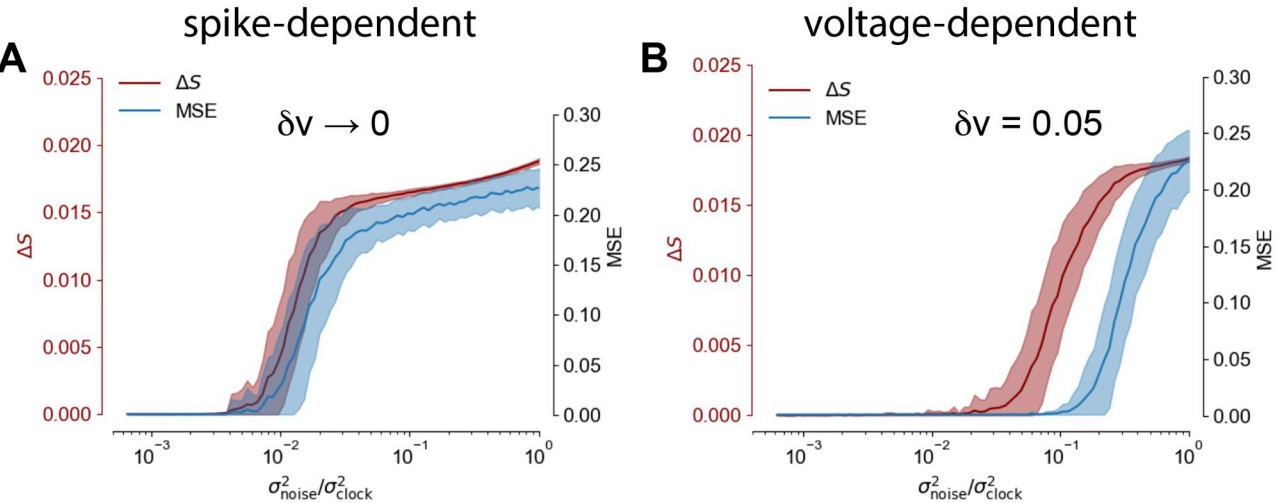

**Fig 2. Robustness to noisy perturbations.** (**A**) (blue) MSE (after 1000 training iterations) as a function of amplitude of the Gaussian noise perturbing the network in the generation mode, $\delta v \rightarrow 0$. Solid line and shaded area: mean and standard deviation of the MSE respectively, statistics evaluated over 50 experiments. (red) Estimation of the number of spikes not following the target pattern in the case $\delta v \rightarrow 0$. (**B**) The same as in panel A for $\delta v = 0.05$.

(example $\sigma^2_{\text{noise}}/\sigma^2_{\text{clock}} = 0.1$) the spontaneous dynamics suffers from significant artifacts, however the network is still able to complete the task with little to no noticeable decrease in performances.

The robustness superiority of the voltage dependent learning rule can be explained as follows. In the case $\delta v \rightarrow 0$, the learning halts as soon as $v$ crosses the threshold appropriately. E.g. if $s_{\text{targ}} = 1$, when $v > 0$ the neuron produces a spike ($s_{\text{pred}} = 1$) and the target is satisfied. Indeed in such case $s_{\text{targ}} - s_{\text{pred}} = 0$, and the weight update is zero (see Eq 6). Vice versa $s_{\text{targ}} = 0$ requires $v < 0$ to get $s_{\text{targ}} - s_{\text{pred}} = 0$. However, since the membrane potential might be very close to the threshold, a small injection of noise can induce an unwanted threshold crossing, thus corrupting the spikes pattern and altering the network prediction. Contrarily, in the voltage dependent formulation the term $s_{\text{targ}} - f(v)$ never vanishes, bringing the neuron further from the threshold and ensuring a safety margin. This process increases the network robustness to noise.

### Learning with a small number of presentations

Fast learning from a small number of examples is a major feature in biological systems. For this reason we show an example in which a trajectory is learned by the system after a very small number of presentations. The task is the same as the one described in the section **Learning 3D trajectories** with the only difference that here the trajectory is composed of 50 time bins.

In Fig 3A each column reports the retrieval of the 3D trajectory after a different number of learning iterations (in particular after 1,2,3 and 4 presentations) when the online learning rule is used. The target sequence is represented in solid lines, while the retrieved trajectory is in dashed lines. After only 4 presentations the trajectory is already learned by the spiking network with a very low error.

Such fast learning is achieved when the online version of the gradient ascent is used.

This approximation has been shown to be a good approximation of the gradient ascent for small learning rate [28]. On the other side, in order to have a fast learning, the learning rate cannot be too small. Here we show that the online approximation, in addition to be local in time, and then biologically plausible, it is extremely beneficial to the fast learning.

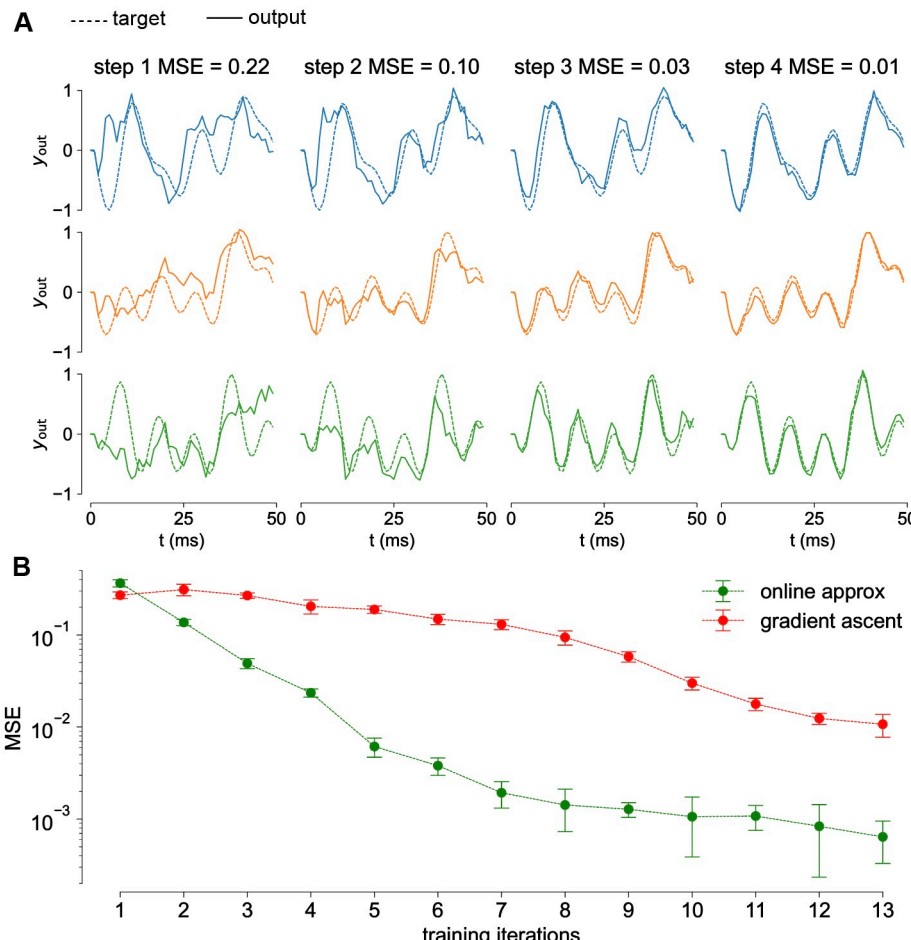

**Fig 3. Learning with a small number of presentations. A)** Columns represent the network output after 1, 2, 3 and 4 learning steps (dashed line: target trajectory, solid line: retrieved trajectory). At the fourth step the network reached an MSE = 0.01. **B)** Online approximation (green) demonstrates to be faster that standard gradient ascent (red). Thick lines and shadings: average and variance over 30 realizations.

We compared the performances of the gradient ascent and its online approximation on this version of the task. The results are reported in Fig 3B respectively in red and green. The online approximation is much faster. It takes on average 5 presentations to learn the trajectory (MSE <0.01), three times less than what required by the standard gradient ascend.

## Walking dynamics

In this task we challenge our proposed framework to address a complex real-world scenario. The quest is to elaborate the problem of learning a realistic multi-dimensional temporal trajectory. We addressed the benchmark problem [15, 37, 38] of learning the walking dynamics of a humanoid skeleton composed of 31 joints (head, femur etc. . .) each of which endowed of rotational degrees of freedom $\boldsymbol{\theta}(t)$. For example the femur can in principle express rotations on three different axes $\theta_{\text{femur}} \in \mathbb{R}^3$, while the wrist is constrained to rotate just around a single axis $\theta_{\text{wrist}} \in \mathbb{R}$. Joints are moreover considered unstretchable, so the set of rotations $\boldsymbol{\Theta} = \{\boldsymbol{\theta}_{\text{joint}}(t)\}$ completely defines the system. The total number of temporal trajectories that our network needs to control is given by the total number of degrees of freedom of the system, which in our

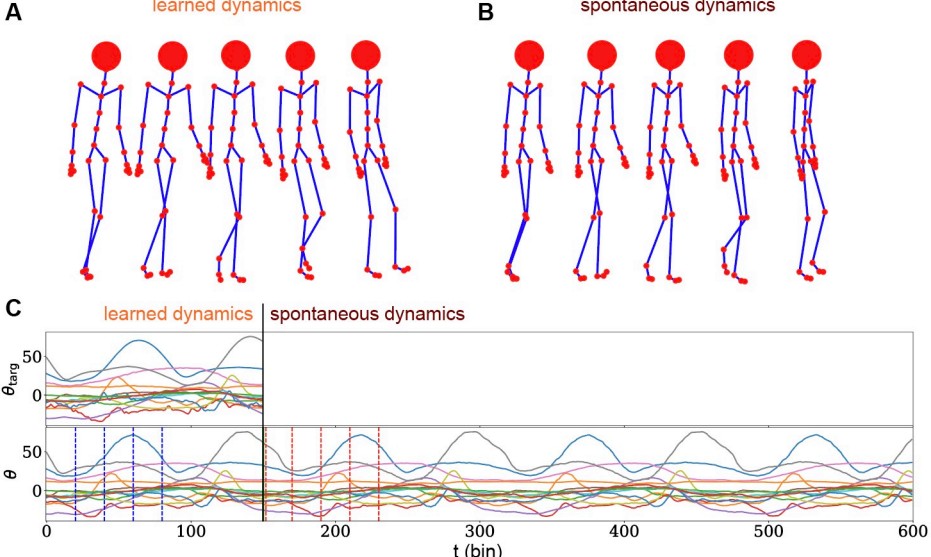

**Fig 4. Walking dynamics. A)** Three-dimensional reconstruction of the walking dynamics produced by the network activity. Reported frames corresponds to $t = \{0, 20, 40, 60, 80\}$. **B)** Three-dimensional reconstruction of the spontaneously generated walking dynamics after the end of the learned-by-memory trajectory. A different, but plausible and exactly periodical behaviour emerges, indicating that the system has successfully learnt how to construct and independent walking cycle. Frames shown corresponds to $t = \{150, 170, 190, 210, 230\}$. **C)** Collection of 15 out of the 56 temporal angular trajectories for both the target motion (upper plot) and learnt dynamics (lower plot). The system receives explicit instructions for $t < 150$ steps, where training is performed. Vertical dotted blue lines highlight the temporal frames reported in panel A. The system then produced a longer, spontaneous dynamics for $t \in [150, 600]$. Vertical red dotted lines highlight the spontaneous activity frames reported in panel B.

problem evaluates to $D = 56$. Data used in this task are obtained from the database developed in [32].

An interesting property of the walking dynamics is indeed its periodicity (data is however derived from motion capture techniques that invalidate exact periodicity). It is thus interesting to explore if the network, trained on a single cycle, is capable of correctly generalize the learnt dynamics by producing novel, plausible, walking behaviour.

The protocol used for this task is exactly the same presented in Learning 3D Trajectories section, with the only difference of the dimension of the trajectory to be learned which is $D = 56$ instead of 3. From this follow that the input matrix $\boldsymbol{J}_{\text{teach}} \in \mathbb{R}^{N \times D}$ and that the output layer is composed of $D$ neurons. During training, the system receives a clock which is identical to what has been described for the 3D trajectories task. During the spontaneous dynamics we simply replicate the training signal in a cyclic fashion. The network parameters for this task are defined in Table 1 for details. Results of the described procedure are reported in Fig 4.

The network successfully memorizes and retrieves the target dynamics with a very small error (MSE = 0.026 on normalized trajectories) after 250 iterations of Adam Optimizer (Fig 4A and 4C up to $t = 150$). Also we let the model generate the trajectory after the end of the learned one. Interestingly the network is capable to generate spontaneously a plausible almost periodic dynamics (Fig 4B and 4C from $t = 150$ on).

## Temporal XOR

In order to further validate the generality of the proposed learning framework, we assessed the well known temporal XOR task. In this complex temporal task the network has to integrate

cues over time and respond at the right moment. The system is asked to respond at time $t_3$ with a non-linear XOR transformation between the bits $A_{in}^1$ and $A_{in}^2$ encoded by two input signals $a_1^t$ and $a_2^t$ provided at preceding times $t_1$ and $t_2$. They encode the desired bit using the length of the duty cycle of a single pulse square wave: (50%→0, 25%→1). This defines 4 possible input signals $x_\mu^t = a_1^t + a_2^t$, $\mu \in \{0, 1, 2, 3\}$. The system target response $\mathbf{y}_{targ}^t$ is a smooth wave form centered at time $t_3$ whose amplitude $A_{out}^3$ encodes the XOR computation of the two inputs: $A_{out}^3 = 2(A_{in}^1 \oplus A_{in}^2 - 0.5)$. This particular choice for the definition of the task is to reproduce the protocol described in [15].

The system is trained to generate four internal target sequences $\mathbf{s}_\mu^{targ}$ on the four possible input combinations. Each sequence is produced by extracting the spontaneous activity of an untrained network receiving as input both the two square-wave signals encoding the bits to be processed $\mathbf{I}_{bits}^t = \mathbf{J}_\mu^{in} \mathbf{x}_\mu^t$ and, similarly to the previous task, the encoded target response $\mathbf{I}_{teach}^t = \mathbf{J}_{teach}^{in} \mathbf{y}_{targ}^t$. We notice that $\mathbf{I}_{bits}^t$ is the equivalent of the clock current described in the previous tasks. In this task such a current is essential since it provides the input signal, which has to be processed by the network. The projecting matrices $\mathbf{J}_\mu^{in}$ and $\mathbf{J}_{teach}^{in}$ are static random Gaussian matrix with zero mean and variance $\sigma_{in}$ and $\sigma_{targ}$. The recurrent network training increases the likelihood of all the four target activities $\mathbf{s}_\mu^{targ}$ by processing one sequence at a time in random order and performing the maximum-likelihood prescribed updating rule. Concurrently a standard linear readout is trained to decode the four sequences $\mathbf{s}_\mu^{targ}$ with MSE objective function and standard Adam optimizer with default parameters.

Fig 5 summarizes the outcome of the training procedure on the temporal XOR task: all the four possible inputs combination are correctly handled by the system, which learns to accurately reproduce the desired signal (see Table 1 for the number of neurons and the other parameters used for this task).

Finally, we mention that we successfully extended the temporal XOR to a temporal parity check, in which more then 2 consecutive inputs are presented in time (see S1 File). We show good results for 3 and 4 consecutive input bits.

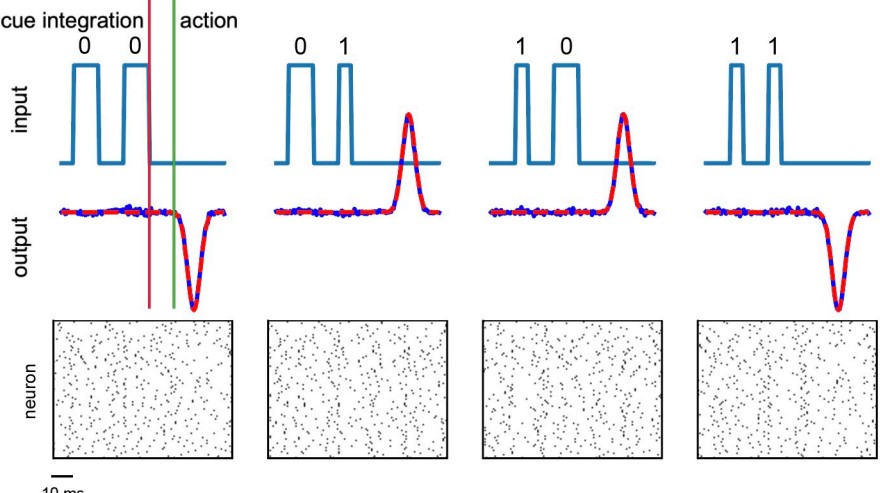

**Fig 5. Results for the temporal XOR test.** The four possible inputs combinations of the binary temporal XOR operation are reported together with the produced output response of the system. (top) On the upper part of each graph the incoming signals are plotted as a function of time, underlining the temporal structure of the task, where the bit is encoded in the duty cycle of a square wave. On the lower part of the graph both the target and the produced network response activity are reported. Continuous blue lines are used for the retrieved output signals, while dashed red lines represents the desired correct output. (bottom) Rastergram of the related internal spiking activity.

## Learning rule for coBa neurons

As mentioned above our approach is general and allows to analytically derive an optimal plasticity rule for different neuronal models. Namely, a different differential equation brings to a different plasticity rule. We already derived such rule for current-based leaky integrate and fire neurons, here we report the learning rule for conductance-based neurons. In this case the evolution for its membrane potential is different because the input affect its dynamics in a multiplicative fashion with the membrane potential $v_i^t$ itself and is described by the following equation:

$$\boldsymbol{v}^t = \quad \left(1 - \frac{\Delta t}{\tau_m}\right)\boldsymbol{v}^{t-1} + \frac{\Delta t}{\tau_m}[\boldsymbol{I}^t + v_{\text{rest}}]$$
$$+ \frac{\Delta t}{\tau_m}\left[(E_{\text{exc}} - \boldsymbol{v}^{t-1})\boldsymbol{J}_{\text{exc}}\hat{\boldsymbol{s}}_{\text{exc}}^{t-1} + (E_{\text{inh}} - \boldsymbol{v}^{t-1})\boldsymbol{J}_{\text{inh}}\hat{\boldsymbol{s}}_{\text{inh}}^{t-1}\right] - J_{\text{res}}\boldsymbol{s}^{t-1} \tag{9}$$

Where $E_{\text{exc}}$ and $E_{\text{inh}}$ are the reversal potentials for excitatory and inhibitory conductances. exc and inh subscripts distinguishes between excitatory and inhibitory neurons and synapses. This means that the populations of neurons has to be segregated in excitatory and inhibitory neurons.

Thus the spike response function is different for excitatory and inhibitory weights and are defined iteratively as

$$\nabla_{J_{\text{exc}}}\boldsymbol{v}^t = \left(1 - \frac{\Delta t}{\tau_m} - \frac{\Delta t}{\tau_m}|\boldsymbol{J}|\,\hat{\boldsymbol{s}}^{t-1}\right)\nabla_{J_{\text{exc}}}\boldsymbol{v}^{t-1} + \frac{\Delta t}{\tau_m}(E_{\text{exc}} - \boldsymbol{v}^{t-1})\hat{\boldsymbol{s}}_{\text{exc}}^{t-1} \tag{10}$$

and

$$\nabla_{J_{\text{inh}}}\boldsymbol{v}^t = \left(1 - \frac{\Delta t}{\tau_m} - \frac{\Delta t}{\tau_m}|\boldsymbol{J}|\,\hat{\boldsymbol{s}}^{t-1}\right)\nabla_{J_{\text{inh}}}\boldsymbol{v}^{t-1} + \frac{\Delta t}{\tau_m}(E_{\text{inh}} - \boldsymbol{v}^{t-1})\hat{\boldsymbol{s}}_{\text{inh}}^{t-1} \tag{11}$$

The segregation of the network in excitatory and inhibitory neurons requires paying attention to the constraint that in the optimization the sign of the synapses has to be conserved. We compared the performances of coBa and cuBa networks finding comparable results (see S1 File for details).

## Discussion

In recent years a wealth of novel training procedures have been proposed for recurrent biological networks, both continuous and spike-based. This work proposes a maximum-likelihood target-based learning framework for recurrent spiking systems. Borrowing from the Machine Learning and in particular Deep Learning community, the aim is to enable learning in complex systems by defining a suitable architecture and an objective function to be optimized, from which the synaptic update rule is derived. Even though biological neurons have quite deterministic dynamics, the modeling of additional noise is required to reproduce effects as synaptic noise and noisy signal coming from other areas of the brain. For this reason the single neuron can be approximated as a stochastic unit whose dynamics is described by a likelihood. The likelihood of a complete network activity is chosen as the training objective function. In this picture the learning in a biological network is understood to be the processes during which the synaptic matrix is adapted to reproduce a target dynamics with high probability. From this simple theoretical assumption explicit synaptic update rules are derived, which offer a clean interpretation of the single terms. This ensures a tight control on the biological realism of the learning rule. The proposed target-based training protocol has been tested on several

temporal tasks: learning to generate trajectory with different dimensionalities (a 3D trajectory and a 56D walking dynamics) and the temporal XOR, which is a non-linear classification problem. In the 3D trajectory generation task the proposed target-based algorithm outperformed the state-of-the-art error-propagation-based algorithms, including e-prop 1 [12] and backpropagation through time. Our model relies on two main assumption: the presence of a teaching signal arriving to the recurrent network and the capability of the neuron to evaluate a local error. The teaching signal is important to generate the internal target representation, and it is plausible to assume that is provided by other cortical areas. The local error is evaluated by the single neuron, and is the difference between the tendency of the neuron to produce a spike and the desired spikes defined by the teaching signal. Theoretical models suggest that this signal might be evaluated by a 2 compartment neuron [34]. Alternately, it has been shown that this learning rule together with a time locality constrain results to be very similar to a standard STDP [26]. Interestingly, the voltage-dependent variant of such rule (see Eq (7)) appears to be coherent with what has been proposed in [35], where the plasticity rule depends on the membrane potential of the postsynaptic neuron.

Different models of neurons lead to different plasticity learning rules. E.g. for conductance based and current based neurons [39, 40] different theoretical learning rules are derived in this work. Such diversity is a theoretical prediction and can be verified experimentally.

## Comparison with other models and biological plausibility

The proposed learning protocol improves the biological plausibility of network architectures used in other recent papers [12, 15, 28]). The model described in [12] relies on a random projection of the errors to the network. On the other hand we bet on a random projection of the desired output into the network, while each neuron evaluate a local error which defines the plasticity rule. It is very intuitive to imagine that every portion of the cerebral cortex is a recurrent neural network receiving signals from the other cortices inducing an internal dynamics during a specific task. Such hypothesis, which reinforces the necessity of supervised learning, is inspired by the so termed 'referent activity templates' which are spike patterns generated by neural circuits present in other portions of the brain, which are to be mimicked by the network subjected to learning [30, 31].

Also, our protocol is computationally efficient since learning requires a calculation of order $O(N^2 \times T \times P)$ where $P$ is the number of presentations of the target pattern. Indeed, the computational time is proportional to the number of synapses, to the trial duration and to the number of repetitions. The online approximation further decreases the computational cost by reducing the number of required presentations $P$.

However, even though our target-based approach is extremely successful in the tasks we presented, we believe that the brain probably uses different learning strategies in different situations. For example the target-based strategies can be a good approach when imitation is required. On the other hand error-based learning can be extremely helpful when a reward is involved.

In the sake of biologically plausibility, in order to achieve a complete locality both in space and in time, we proposed what we called the online approximation, which is an update of the weights at every time $t$, where only the information available at the current time is used. Namely it is not necessary to wait for the end of the presentation of training example to perform the weight update.

Models for fast learning have been proposed, and in general they rely on a suited pre-training or preparation of the recurrent network [12, 41–43]. This procedure is not task specific. Subsequently, the training on the task is performed. We do not investigate an optimization of

the network before the learning task. However, we find that our time local approximation is extremely advantageous when learning from a small number of examples, increasing significantly the learning velocity.

We remark that the analytic formulation used to derive our learning rule resembles those proposed in [26, 28, 34], but they mainly focused on training the network to learn specific temporal patterns of spike. Here we integrate such approach with a target-based learning protocol, allowing to solve complex temporal task.

This choice is similar to the one described in the full-FORCE scheme [14, 15]. Although similar, they are not equivalent, indeed the full-FORCE learning algorithm aims to reproduce the target input current to each neuron. On the other hand our target is a specific spatio-temporal spike pattern, allowing for precise spike timing coding. Such findings taken together suggest that the capability of a neuron to estimate a local error is beneficial to the learning process. Experimental evidences suggest that single neurons are indeed capable to integrate different stimuli to estimate local errors [33]. In addition to this in [34] it is described a plausible mechanism to achieve such an error in a two-compartment neuron model.

Furthermore, in our model, the weights are optimized in order to reproduce a specific spatio-temporal pattern of spikes, a relevant feature in biological neural networks (see Introduction). Also, this allows us to reproduce target output trajectories with a tremendous precision. A target spatio-temporal spike pattern is not directly implied in the full-FORCE, where the total input current received by each from other neurons is considered as a target function.

Finally, we propose solutions that further improve the biological plausibility of the model, such as the online approximation and the online evaluation of the target sequence within the neuron.

## Methods

### Theoretical derivation of the learning rule for cuBa neurons

In our formalism neurons are modeled as real-valued variable $v_j^t \in \mathbb{R}$, where the $j \in \{1, \ldots, N\}$ label identifies the neuron and $t \in \{1, \ldots, T\}$ is a discrete time variable. Each neuron exposes an observable state $s_j^t \in \{0, 1\}$, which represents the occurrence of a spike from neuron $j$ at time $t$.

We then define the following probabilistic dynamics for our model:

$$\hat{\boldsymbol{s}}^t = \left(1 - \frac{\Delta t}{\tau_s}\right)\hat{\boldsymbol{s}}^{t-1} + \frac{\Delta t}{\tau_s}\boldsymbol{s}^t$$

$$\boldsymbol{v}^t = \left(1 - \frac{\Delta t}{\tau_m}\right)\boldsymbol{v}^{t-1} + \frac{\Delta t}{\tau_m}(\boldsymbol{J}\hat{\boldsymbol{s}}^{t-1} + \boldsymbol{I}^t + v_{\text{rest}}) - J_{res}\boldsymbol{s}^{t-1}$$

$$p(s_i^{t+1}|v_i^t) = \frac{\exp\left[s_i^{t+1}\left(\frac{v_i^t - v_i^{\text{th}}}{\delta v}\right)\right]}{1 + \exp\left(\frac{v_i^t - v_i^{\text{th}}}{\delta v}\right)}$$

Where $\Delta t$ is the discrete time-integration step, while $\tau_s$ and $\tau_m$ are respectively the spike-filtering time constant and the temporal membrane constant. Each neuron is a leaky integrator with a recurrent filtered input obtained via a synaptic matrix $\boldsymbol{J} \in \mathbb{R}^{N \times N}$ and an external signal $\boldsymbol{I}^t$. $v^{th}$ is the firing threshold and $\delta v$ defines the amount of noise in the spike generation. $J_{res}$ accounts for the reset of the membrane potential after the emission of a spike.

In the $\delta v \to 0$ limit the generation is deterministic and $p(s_i^{t+1}|v_i^t) = \Theta[s_i^{t+1}(v_i^t - v^{th})]$. The log-likelihood of a complete network activity $s$ can be expressed as:

$$\mathcal{L}(s;J) = \log p(s;J) = \log \prod_{t=1}^{T} \prod_{i=1}^{N} p(s_i^{t+1}|v^t;J) = \sum_{t=1}^{T} \sum_{i=1}^{N} \log p(s_i^{t+1}|v^t;J)$$

Where we assumed the parallel update of all the neurons at each time step, and that the probabilistic generation of the spike is independent between neurons at the same time step. Indeed our noisy generation is equivalent to the presence of a source of noise not explicitly described in the system. A possible example is an external source of noise for each neuron. Our description doesn't account for correlated sources of noise.

We notice that in order to evaluate such likelihood it is important to be able to evaluate the membrane potential at each time $t$. In order to do so it is necessary to know the spikes $s^t$ produced by the network, the law of evolution of the membrane potential (see Eq (4)) and its initial condition $v^{t=1}$ (which we usually define as a constant $v^0$).

The idea now is to exploit the introduced likelihood $\mathcal{L}(s;J)$ as a valuable tool for a target-based learning. We introduce a target activity $s_{\text{targ}}$ and exploit the dependence of the likelihood on the recurrent weights $J$ to increase the likelihood of observing the target pattern as the system's spontaneous activity. In particular we compute:

$$\nabla_J \mathcal{L}(s_{\text{targ}};J) = \sum_{t=1}^{T} \nabla_J \log p(s_{\text{targ}}^t|v^{t-1};J) \tag{12}$$

were $v^{t-1}$ is in turn a function of $s_{\text{targ}}$. Indeed, it is computed using Eq (4) and replacing $s$ with $s_{\text{targ}}$. This framework thus prescribes to implement as a viable learning rule the likelihood gradient $\nabla_J \mathcal{L}$. Via this optimization protocol, the system learns to exploit its resources to encode the desired activity.

The maximum likelihood learning rule then prescribes:

$$\frac{\partial \mathcal{L}}{\partial J_{ik}} = \frac{1}{\delta v} \sum_{t=1}^{T} \sum_j \left[ s_{j,\text{targ}}^{t+1} - \frac{\exp \frac{v_j^t - v^{th}}{\delta v}}{1 + \exp \frac{v_j^t - v^{th}}{\delta v}} \right] \frac{\partial v_j^t}{\partial J_{ik}} \tag{13}$$

Where in Eq (13) we have rewritten the likelihood gradient using the index notation. The last term $\frac{\partial}{\partial J_{ik}} v_j^t$ can be iteratively written by differentiating Eq (4):

$$\frac{\partial v_j^{t+1}}{\partial J_{ik}} = \left(1 - \frac{\Delta t}{\tau_m}\right) \frac{\partial v_j^t}{\partial J_{ik}} + \frac{\Delta t}{\tau_m} \hat{s}_{k,\text{targ}}^t \delta_{ij} \tag{14}$$

and setting an initial condition, e.g. $\frac{\partial v_j^{t=1}}{\partial J_{ik}} = 0$. We stress that the differential operator $\nabla_J$ is only applied to $v^{t-1}$ and not to $\hat{s}_{\text{targ}}^{t-1}$, because the latter represents the desired target dynamics, which is assumed to be fixed throughout the training process and thus expresses no dependence on the synaptic matrix $J$.

The use of $s_{\text{targ}}^t$ in the pre-synaptic term is a consequence of likelihood maximization. However in [29], it as been proposed to replace $s_{\text{targ}}^t$ with the activity generated by the network during the trial, proving that this induces a small error in the learning protocol. Here we claim the biological plausibility to keep $s_{\text{targ}}^t$ in the pre-synaptic term because of the presence of the dedicated apical compartment, which make accessible to the network the target pattern of spikes.

Because of the Kronecher $\delta$ in Eq (14), together with its initial condition, $\frac{\partial v_j^t}{\partial J_{ik}}$ differs from zero only when $j = i$. We can thus finally write

$$\frac{\partial \mathcal{L}}{\partial J_{ik}} = \frac{1}{\delta v} \sum_{t=1}^{T} \left[ s_{i,\text{targ}}^{t+1} - \frac{\exp \frac{v_i^t - v^{\text{th}}}{\delta v}}{1 + \exp \frac{v_i^t - v^{\text{th}}}{\delta v}} \right] \frac{\partial v_i^t}{\partial J_{ik}} \tag{15}$$

It follows that the weight plasticity rule can be expressed as

$$\begin{aligned}
\Delta \boldsymbol{J} &= \frac{\eta}{\delta v} \sum_{t=1}^{T} \left[ \boldsymbol{s}_{\text{targ}}^{t+1} - \frac{\exp \frac{\boldsymbol{v}^t - \boldsymbol{v}^{\text{th}}}{\delta v}}{1 + \exp \frac{\boldsymbol{v}^t - \boldsymbol{v}^{\text{th}}}{\delta v}} \right]^{\top} \nabla_J \boldsymbol{v}^t \\
&= \eta_0 \sum_{t=1}^{T} [\boldsymbol{s}_{\text{targ}}^{t+1} - f(\boldsymbol{v}^t)]^{\top} \nabla_J \boldsymbol{v}^t
\end{aligned} \tag{16}$$

Where we defined $f(\boldsymbol{v}^{\text{th}}) = \frac{\exp \frac{\boldsymbol{v}^t - \boldsymbol{v}^{\text{th}}}{\delta v}}{1 + \exp \frac{\boldsymbol{v}^t - \boldsymbol{v}^{\text{th}}}{\delta v}}$. We observe that this learning rule is voltage dependent since $f(\boldsymbol{v}^{\text{th}})$ is a function of the membrane potential $\boldsymbol{v}^{\text{th}}$.

It is possible to rewrite this expression by recognizing that the second term in the first factor, in the limit $\delta v \to 0$ and with $\eta_0 = \eta/\delta v$ finite, represents the network prediction for the spike vector $\boldsymbol{s}_{\text{pred}}^{t+1}$ value at the subsequent time step, based on the current network hidden state $\boldsymbol{v}^t$. Shifting the summed index $t$ one obtains:

$$\Delta \boldsymbol{J} = \frac{\eta}{\delta v} \sum_{t=0}^{T-1} [\boldsymbol{s}_{\text{targ}}^{t+1} - \boldsymbol{s}_{\text{pred}}^{t+1}]^{\top} \nabla_J \boldsymbol{v}^t \tag{17}$$

This version of the learning rule is no longer voltage dependent and we refer to it as spike-dependent. We note how the obtained expression offers a simple interpretation: it effectively separates into a first learning signal, which is the neuron-wise difference between the teaching signal and the spontaneous network-induced activity, and an eligibility trace.

Using this maximum-likelihood framework we have obtained an explicit expression for the synaptic weight update, a result that previously eluded other target-based learning algorithm [14].

## Online approximation

The gradient of the likelihood formulated in Eq 6 requires to accumulate the weight updates over the all training trial. We performed an online approximation by removing the sum over time and updating the weights at every time step.

$$\Delta J_{ik}^t(\{J_{ik}^{1,\dots t}\}) = \eta_0 \left[ s_{i,\text{targ}}^{t+1} - s_{i,\text{pred}}^{t+1}(\{J_{ik}^{1,\dots t}\}) \right] \frac{\partial v_i^t}{\partial J_{ik}} \tag{18}$$

and the total weights update after the whole trial is

$$\Delta J_{ik} = \sum_t \Delta J_{ik}^t(\{J_{ik}^{1,\dots t}\}) \tag{19}$$

In Eq (18) we are making explicit that at every time step the weights are different, and the update at a specific time step depends on the weights at the same time (and at the previous times).

$s_{i,\text{pred}}^{t+1}$ in Eq (18) is the only term depending on the weights $J_{ij}^t$. This means that the gradient ascent and the online approximation are equivalent with the only difference that in the online case the weights, and then the prediction, are updated at every time. The same stands for the voltage-dependent rule.

### Generation mode

When the network performs the task it is set in generation mode. The plasticity is turned off and the network is initialized with the proper initial conditions $v^{t=1} = v^0$ (which is the same as the one defined in the likelihood maximization protocol). We also investigate the case of noisy initial conditions (see S6 Fig in S1 File). The the voltage and the spiking dynamics follow respectively Eq (4) and the deterministic limit ($\delta v \to 0$) of Eq (2).

### Readout training

The training of the readout weights is performed through a standard minimization of the MSE between the output $y = J_{\text{out}} \hat{s}_{out}$ and the target output $y_{\text{targ}}$, which results in the following rule

$$\Delta J_{\text{out}} = [y_{\text{targ}} - J_{\text{out}} \hat{s}_{out}] \hat{s}_{out}^{\top} \tag{20}$$

Where $Y_{\text{targ}} = \{y_{\text{targ}}^t\}$, $Y_{\text{targ}} \in \mathbb{R}^{3 \times T}$ is the matrix that collects the target signal over time. $s \in \mathbb{R}^{N \times T}$ is the matrix that collects the spikes emitted by the network over time and $\hat{s}_{out}$ is its exponential temporal filtering with a time scale $\tau_{out}$. (similarly to Eq (5)). See Table 1.

When the online approximation is used to train the network also the the readout is updated at every time step as follows

$$\Delta J_{\text{out}} = [y_{\text{targ}}^t - J_{\text{out}} \hat{s}_{out}^t] \hat{s}_{out}^{t,\top} \tag{21}$$

### Simulation parameters and source code

The source code is available for download under CC-BY license in the https://github.com/myscience/LTTS public repository. We report in the table below the network parameters used in the different tasks.

In the 3D trajectory benchmark we used realistic synaptic and membrane timescales, in order to show that we achieved good results with biologically plausible parameters. A smaller membrane timescale usually facilitates the convergence of the method, for this reason we used a shorter time scale in order to further decrease the number of steps required for the learning.

### Supporting information

**S1 File.**
(PDF)

**S1 Video.**
(ZIP)

## Author Contributions

**Conceptualization:** Paolo Muratore, Cristiano Capone, Pier Stanislao Paolucci.

**Data curation:** Paolo Muratore, Cristiano Capone.

**Formal analysis:** Paolo Muratore, Cristiano Capone.

**Funding acquisition:** Pier Stanislao Paolucci.

**Investigation:** Cristiano Capone.

**Methodology:** Paolo Muratore, Cristiano Capone.

**Project administration:** Pier Stanislao Paolucci.

**Resources:** Pier Stanislao Paolucci.

**Software:** Paolo Muratore, Cristiano Capone.

**Supervision:** Pier Stanislao Paolucci.

**Validation:** Paolo Muratore, Cristiano Capone, Pier Stanislao Paolucci.

**Visualization:** Paolo Muratore, Cristiano Capone.

**Writing – original draft:** Paolo Muratore, Cristiano Capone.

**Writing – review & editing:** Paolo Muratore, Cristiano Capone, Pier Stanislao Paolucci.

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
