## [Decision Letter · Decision Letter 0]

22 Jul 2020

PONE-D-20-15736

Target spiking patterns enable efficient and biologically plausible learning for complex temporal tasks

PLOS ONE

Dear Dr. Muratore,

Thank you for submitting your manuscript to PLOS ONE. After careful consideration, we feel that it has merit but does not fully meet PLOS ONE’s publication criteria as it currently stands. Therefore, we invite you to submit a revised version of the manuscript that addresses the points raised during the review process.

The manuscript should be revised for clarity and address concerns of the reviewers  about details of the algorithm. Suggested literature should be discussed. The following specific concerns should be addressed.

The Authors claim to use leaky integrate and fire neurons (see Eq. (3)), but in the discussion no reset of v_t is mentioned. Such resets, while being a potentially essential source of non-linearity in the dynamics of the network, could be problematic when calculating derivatives of v_t with respect to J. Please, discuss this issue.

The Authors claim that their method is biologically plausible. However, equation (16) is recursive and requires the storage of the full history of membrane potentials and spikes throughout the training. This does not seem to be biologically plausible (two phases needed, and how are the values stored in the biological substrate?) and may be impractical in engineering applications. It is not clear whether the online version of the algorithm given by (6) alleviates these problems, because not enough details of this procedure are currently given.  For example, are targets still generated during a separate phase in the online training paradigm?

-Please, clarify how the 'internal target spike trains' are obtained. How are the spike trains obtained from a teaching input current I_teach?

-The teaching current is projected randomly to all the neurons in the network. What if some neurons do not receive a teaching current? In Brea et al. there is a pool of hidden neurons for example. Is it possible to only train the read-out weights for the 3D trajectory?

-It is claimed that spike-timing is a crucial feature of the presented model. Please, explain how the tasks (3D trajectories, walking dynamics and XOR) are related to exact spike timing in the RNN. Especially since the read-out seems to work based on filtered spike trains from the RNN.

-At the end of the discussion the authors state: 'A major difference is that our learning method relies on optimizing the timing of the spikes emitted by each neuron rather that optimizing the time course of their input current.' -Is the model robust to perturbations?

-The cartoon of the model is not readable, maybe due to the pdf conversion of the figure.

-Line 137: what is K? What does the clock input look like? A figure would help here.

-Line 267: other neuron models, can you show an implementation? Maybe a figure comparing performance of the two different neuron models?

Some statements are made without explaining them, or citing literature.

Line 80: ‘makes the learning extremely faster’ why?

Line 178: clock is not necessary, show this? Why?

Line 197: reference?

Line 330: where does this N^2 x T x P come from?

Line 424: reference? Why is the membrane potential related to the learning speed?

Table 1: Why is the membrane potential for the online approximation four times smaller than the other tasks (2ms vs 8ms)?

If indeed this is related to learning speed it could explain why the online version of the learning rule is faster.

Could you make the code available for the reviewers?

Eq. (1): Left-hand side does not depend on v, right-hand side does. The same problem appears in (13) and (14). It would be beneficial for the reader to explain this here.

Check  the time indices in (16).

113; "We refer to such likelihood as L(s_targ ; J ) where J are the recurrent synaptic weights of the system":  (log)-likelihood was already introduced in (1), so this is a bit confusing.

188; What are "peculiar rotational degrees of freedom"?

As mentioned the algorithm is very similar to full-FORCE. Perhaps a more detailed analysis of similarities and differences would be useful?

Some claims should be supported by references, e.g.,  before equation (7) (is it a novel equation, or is it based on the literature?), "the human brain works with only 20 watt".

Are neurons or synapses conductance- or current-based? Why are connections segregated into excitatory and inhibitory in the case of coBa, but not cuBa? Why no in-silico experiments were performed with coBa?

The "generation mode" is not very biologically plausible: the neural network is always set in the same initial condition. To be more realistic the network should start from random initial conditions and be cued by the input.

It is not clear whether the reported performance results are typical, average (over many training instances) or cherry-picked.

"Learning to generate 3D trajectories". In [Bellec et al, 2019] they use random amplitude drawn from [0.5, 2.0] range. Is 2.5 a typo here?

The conditions should be the same for correct method comparisons.

Algorithm performance comparison. A confidence interval should be provided for the method's MSE

(despite the fact that Bellect et al don't provide such data).

The protocol can be run for different inputs to collect statistics.

It would be nice to illustrate faster convergence of new protocol in a plot

together with the data from Bellec et al.

 "Online approximation" is used only for "few representations" task ?

If so, it should be explained why it doesn't work well for the other tasks if it helps faster learning.

The citation for "Diederik P. Kingma and Jimmy Lei Ba. Adam : A method for stochastic optimization. 2014. arXiv:1412.6980v9" is needed.

Spelling and grammatical errors:

Abstract ...require very few examples.

Line 53: ‘what is called’ This sentence might benefit from a rewrite.

Line 87: p(s(t)|v(t)) = .. spiking dynamics at time t is a function of membrane potential at time t, in equation 2 however it is the spiking dynamics at time t+1. Is this a mistake or something I missed?

Line 108: ‘the spikes undergo’

53: what is call -> what is called

74: vice versa

108: spikes undergoes -> spikes undergo

132: relationship

Fig 1 caption :

... can be though -> can be thought

... the different -> the difference

... It is thus shown how our algorithm ... -> It is thus shown that our algorithm ...

164: turned off

Fig 3 caption: Rastergam -> Rastergram

Formula 6:  the last symbol alignment

271: the learning rule

276: distinguishes

282: requires paying attention

293: the learning

337: is inspire (?)

340: used to derive (?)

341: double [20] citation

344: full-FORCE

346: on the other hand

349: a misplaced citation - suggests[25]

We look forward to receiving your revised manuscript.

Kind regards,

Gennady Cymbalyuk, Ph.D.

Academic Editor

PLOS ONE

Journal Requirements:

Reviewers' comments:

Reviewer's Responses to Questions

**Comments to the Author**

1. Is the manuscript technically sound, and do the data support the conclusions?

Reviewer #1: Partly

Reviewer #2: Partly

Reviewer #3: Yes

2. Has the statistical analysis been performed appropriately and rigorously? 

Reviewer #1: N/A

Reviewer #2: I Don't Know

Reviewer #3: No

3. Have the authors made all data underlying the findings in their manuscript fully available?

Reviewer #1: No

Reviewer #2: No

Reviewer #3: No

4. Is the manuscript presented in an intelligible fashion and written in standard English?

Reviewer #1: Yes

Reviewer #2: No

Reviewer #3: Yes

5. Review Comments to the Author

Reviewer #1: The manuscript titled ‘Target spiking patterns enable efficient and biologically plausible learning for complex temporal tasks’ by Muratore et al. contains an interesting proposal. By providing a target signal to the neurons in an RNN they derive a learning rule that changes the parameters of the network to match the target signal. Most interestingly they show that the non-trivial temporal xor task can be learned. While potentially relevant for the study of spiking networks and temporal behaviour, some points have to be clarified and extended before publication can be recommended. Below I summarize the main points.

1. Novelty

-The study of Brea et al. (Journal of neuroscience 2013) seems very closely related to this study. In fact, the derivation of the learning rule for the specific dynamics is almost the same. It would be helpful to discuss the differences/advances made in this study.

-Fast and accurate learning of tasks has been proposed recently in various models, for example in Klos et al. (arXiv 2019) and Nicola and Clopath (Nat neuro 2019). It would be helpful to understand how this work relates to theirs.

-Some models based on realistic plasticity (stdp-like) and dynamics have recently been proposed to learn storage and recall of sequences. A few examples are Murray and Escola (eLife 2017), Maes et al. (plos cb 2020), Cone and Shouval (arXiv 2020). To what extent are the ideas in these papers similar? For example also in these models a target input is provided and learned (in a few different ways).

2. Training

-It is not clear to me how the 'internal target spike trains' are obtained. I can see that there is a teaching input current I_teach present, but how are the spike trains obtained from this? I assume that they are recorded during spontaneous dynamics and afterwards in a training phase are used as a target where the I_teach is not present anymore.

-The teaching current is projected randomly to all the neurons in the network. What if some neurons do not receive a teaching current? In Brea et al. there is a pool of hidden neurons for example. Is it possible to only train the read-out weights for the 3D trajectory?

-It is claimed that spike-timing is a crucial feature of the presented model. However, I do not understand how the tasks (3D trajectories, walking dynamics and XOR) are related to exact spike timing in the RNN. Especially since the read-out seems to work based on filtered spike trains from the RNN.

-At the end of the discussion the authors state: 'A major difference is that our learning method relies on optimizing the timing of the spikes emitted by each neuron rather that optimizing the time course of their input current.' My question here is similar to what I wrote above: I am not sure how this works.

-Is the model robust to perturbations?

3. Graphical presentation of results

-The cartoon of the model is not readable, maybe due to the pdf conversion of the figure.

-Line 137: what is K? What does the clock input look like? A figure would help here.

-Line 267: other neuron models, can you show an implementation? Maybe a figure comparing performance of the two different neuron models?

4. Explanations

Some statements are made without explaining them, or citing literature.

Line 80: ‘makes the learning extremely faster’ why?

Line 178: clock is not necessary, show this? Why?

Line 197: reference?

Line 330: where does this N^2 x T x P come from?

Line 424: reference? Why is the membrane potential related to the learning speed?

Table 1: Why is the membrane potential for the online approximation four times smaller than the other tasks (2ms vs 8ms)?

If indeed this is related to learning speed it could explain why the online version of the learning rule is faster.

5. Language errors

Throughout the manuscript there are both spelling and grammatical errors. A few of those are listed here:

Abstract ...require very few examples.

Line 53: ‘what is called’ This sentence might benefit from a rewrite.

Line 87: p(s(t)|v(t)) = .. spiking dynamics at time t is a function of membrane potential at time t, in equation 2 however it is the spiking dynamics at time t+1. Is this a mistake or something I missed?

Line 108: ‘the spikes undergo’

6. Code

Is it possible that the code is made available for the reviewers?

Reviewer #2: The manuscript introduces a simple training procedure for spiking neural networks based on two main ingredients: (a) outputs of the network are generated via trainable linear readout, and (b) dynamics of the network depends on the connectivity matrix, which is trained to generate target spike sequence by means of the standard maximization of the likelihood. Target activities are generated during a separate phase in which neurons are driven by inputs and fixed random linear feedback that encodes the desired outputs. The algorithm is tested in-silico on multiple tasks that show its efficiency.

Important issues:

The Authors claim to use leaky integrate and fire neurons (see Eq. (3)), but in the discussion no reset of v_t is mentioned. Such resets, while being a potentially essential source of non-linearity in the dynamics of the network, could be problematic when calculating derivatives of v_t with respect to J. Unfortunately this issue is not discussed.

The Authors claim that their method is biologically plausible. However, equation (16) is recursive and requires the storage of the full history of membrane potentials and spikes throughout the training. This does not seem to be biologically plausible (two phases needed, and how are the values stored in the biological substrate?) and may be impractical in engineering applications. It is not clear to me whether the online version of the algorithm given by (6) alleviates these problems, because not enough details of this procedure are currently given. For example, are targets still generated during a separate phase in the online training paradigm?

In order to properly place this work in the context of current research it would be beneficial for the reader to list other target-based learning schemes, e.g., [1] where targets are provided via auto-encoders, [2] where targets are generated by a chaotic neural network. Although in the manuscript the same network is first used to generate the target activity and then being trained, the target activity is fixed during training (i.e., second phase, see the discussion after (16)). If I understand correctly, recurrent connections are not used in the first phase. One could thus consider the target-generating network as distinct from the trained network, similarly to [2]. Another related work is [3], where it is shown that backpropagation-like algorithm works even if the feedback matrix is random and fixed. Since in the current work target activities are generated via random and fixed linear feedback, this seems relevant. Unfortunately, although this work is cited, this analogy is not discussed. Moreover, such a random feedback was used to train recurrent neural networks in [4]. Another related work is [5].

The manuscript contains numerous language errors and in some places is hard to understand.

Minor issues:

The Authors claim that the Python code with simulations will be available on GitHub. Unfortunately, it does not seem to be available now, so I could not check the code.

Eq. (1): Left-hand side does not depend on v, right-hand side does. The same problem appears in (13) and (14). It would be beneficial for the reader to explain this here. I am assuming that the explanation is related to the fact that v in turn depends on the history of s (assuming some specified initial condition of v).

There is probably a mistake in the time indices in (16).

113; "We refer to such likelihood as L(s_targ ; J ) where J are the recurrent synaptic weights of the system":  (log)-likelihood was already introduced in (1), so this is a bit confusing.

118; I see nothing peculiar about the expression (5). In fact, it looks rather simple, and follows the well known form of delta learning rule.

188; What are "peculiar rotational degrees of freedom"?

As mentioned the algorithm is very similar to full-FORCE. Perhaps a more detailed analysis of similarities and differences would be useful?

Some claims should be supported by references, e.g., before equation (7) (is it a novel equation, or is it based on the literature?), "the human brain works with only 20 watt".

Are neurons or synapses conductance- or current-based? Why are connections segregated into excitatory and inhibitory in the case of coBa, but not cuBa? Why no in-silico experiments were performed with coBa?

The "generation mode" is not very biologically plausible: the neural network is always set in the same initial condition. To be more realistic the network should start from random initial conditions and be cued by the input.

It is not clear whether the reported performance results are typical, average (over many training instances) or cherry-picked.

[1] Dong-Hyun Lee, Saizheng Zhang, Asja Fischer and Yoshua Bengio, Difference Target Propagation, in Machine Learning and Knowledge Discovery in Databases, pages 498-515, Springer International Publishing, 2015

[2] Laje R, Buonomano DV. Robust timing and motor patterns by taming chaos in recurrent neural networks. Nat. Neurosci. 2013;16: 925–933. pmid:23708144

[3] Lillicrap, T., Cownden, D., Tweed, D. et al. Random synaptic feedback weights support error backpropagation for deep learning. Nat Commun 7, 13276 (2016)

[4] Murray, James M. Local online learning in recurrent networks with random feedback. eLife 8 (2019): e43299.

[5] Kim, Christopher M., and Carson C. Chow. Learning recurrent dynamics in spiking networks. eLife 7 (2018): e37124.

Reviewer #3: The authors propose a protocol for training recurrent spiking neural networks(RSNN) without using direct error back propagation.

The protocol uses targed-based learning approach and closely related to full-FORCE training method

described in DePasquale et al, 2018.

The network is trained to recapitulate the activity induced during the "pre-training" phase, when the target signal is provided as the input.

The proposed approach is biologically plausible and provides interesting insights into possible mechanisms

of biological neural networks learning. The references to 2 compartment neuron models are especially interesting.

According to my assessment, the authors need to address the minor points listed below.

1. "Learning to generate 3D trajectories". In [Bellec et al, 2019] they use random amplitude drawn from [0.5, 2.0] range. Is 2.5 a typo here?

The conditions should be the same for correct method comparisons.

2. Algorithm performance comparison. A confidence interval should be provided for the method's MSE

(despite the fact that Bellect et al don't provide such data).

The protocol can be run for different inputs to collect statistics.

3. It would be nice to illustrate faster convergence of new protocol in a plot

together with the data from Bellec et al.

4. "Online approximation" is used only for "few representations" task ?

If so, it should be explained why it doesn't work well for the other tasks if it helps faster learning.

5. The citation for "Diederik P. Kingma and Jimmy Lei Ba. Adam : A method for stochastic optimization. 2014. arXiv:1412.6980v9" is needed.

Typos

53: what is call -> what is called

74: vice versa

108: spikes undergoes -> spikes undergo

132: relationship

Fig 1 caption :

... can be though -> can be thought

... the different -> the difference

... It is thus shown how our algorithm ... -> It is thus shown that our algorithm ...

164: turned off

Fig 3 caption:

Rastergam -> Rastergram

Formula 6: the last symbol alignment

271: the learning rule

276: distinguishes

282: requires paying attention

293: the learning

337: is inspire (?)

340: used to derive (?)

341: double [20] citation

344: full-FORCE

346: on the other hand

349: a misplaced citation - suggests[25]

6. PLOS authors have the option to publish the peer review history of their article (what does this mean?). If published, this will include your full peer review and any attached files.

Reviewer #1: No

Reviewer #2: No

Reviewer #3: **Yes: **Alexander Ivanenko

---

## [Author Response · Author response to Decision Letter 0]

9 Oct 2020

All Editor and Reviewer comments are addressed in the "Letter to Reviewers" which is attached to this revised submission

---

## [Decision Letter · Decision Letter 1]

12 Nov 2020

PONE-D-20-15736R1

Target spike patterns enable efficient and biologically plausible learning for complex temporal tasks

PLOS ONE

Dear Dr. Muratore,

Thank you for submitting your manuscript to PLOS ONE. After careful consideration, we feel that it has merit but does not fully meet PLOS ONE’s publication criteria as it currently stands. Therefore, we invite you to submit a revised version of the manuscript that addresses the points raised during the review process.

Please, check whether during training s_pred and v at time t are calculated assuming that at times t_past<t according="" all="" authors="" been="" between="" consistency="" do="" dynamics="" ensure="" generated="" have="" how="" of="" pattern="" spikes="" spikes.="" target="" the="" to="" v="">?</t>

-Is it possible to make a statement about the capacity of the model? For example, the temporal xor task shows that at least 4 different inputs can be stored in a network of 500 neurons, how many more can be stored?

-It would be nice to mention full-FORCE approach in the introduction in more explicit manner, since that is the most closely related algorithm published so far.

 Not all of the source code is provided, e.g. coBa model is missing

- It is well-known that in the studied scenario minimizing KL divergence is equivalent to maximizing likelihood. Why do you contrast these two approaches?

-- "Statistics is evaluated over 50 realizations for each σ noise value. (25 different 3D trajectory times 25 realizations of the noisy input.)": This is confusing. 25 times 25 equals 50?

-- Equations (12), (13), and (14), (15) are exactly the same as, respectively, (5), (4) and (2), (1). Perhaps such a repetition is not needed?

Language:

-- "In addition, our approach results quite natural in terms of biological plausibility.": ungrammatical

-- "combined to a deterministic generation of the spikes": combined  compared ?

-- "bringing the neuron further from the threshold and ensuring a safety gap": also called margin

line 205 parenthesis

line 312 "The in the case"

We look forward to receiving your revised manuscript.

Kind regards,

Gennady Cymbalyuk, Ph.D.

Academic Editor

PLOS ONE

Reviewers' comments:

Reviewer's Responses to Questions

**Comments to the Author**

1. If the authors have adequately addressed your comments raised in a previous round of review and you feel that this manuscript is now acceptable for publication, you may indicate that here to bypass the “Comments to the Author” section, enter your conflict of interest statement in the “Confidential to Editor” section, and submit your "Accept" recommendation.

Reviewer #1: (No Response)

Reviewer #2: All comments have been addressed

Reviewer #3: All comments have been addressed

2. Is the manuscript technically sound, and do the data support the conclusions?

Reviewer #1: Yes

Reviewer #2: Yes

Reviewer #3: Yes

3. Has the statistical analysis been performed appropriately and rigorously? 

Reviewer #1: N/A

Reviewer #2: Yes

Reviewer #3: Yes

4. Have the authors made all data underlying the findings in their manuscript fully available?

Reviewer #1: Yes

Reviewer #2: Yes

Reviewer #3: No

5. Is the manuscript presented in an intelligible fashion and written in standard English?

Reviewer #1: Yes

Reviewer #2: Yes

Reviewer #3: Yes

6. Review Comments to the Author

Reviewer #1: The manuscript has been improved greatly.

One question I would like to ask is whether it is possible to make a statement about the capacity of the model. For example, the temporal xor task shows that at least 4 different inputs can be stored in a network of 500 neurons, how many more can be stored?

Reviewer #2: In my opinion the Authors have properly addressed most issues raised by me and other Referees. The code in Python is written clearly and is easily accessible through GitHub. Altogether the current version of the manuscript merits its publications in PLOS One. However, I found some other issues that should be addressed before the manuscript is published:

Potentially important:

-- If I understood correctly, during training s_pred and v at time t are calculated assuming that at times t_past<t according="" all="" authors="" been="" between="" consistency="" do="" dynamics="" ensure="" generated="" have="" how="" of="" pattern="" spikes="" spikes.="" target="" the="" to="" v="">

Minor:

-- It is well-known that in the studied scenario minimizing KL divergence is equivalent to maximizing likelihood. I am not sure why the Authors contrast these two approaches.

-- "Statistics is evaluated over 50 realizations for each σ noise value. (25 different 3D trajectory times 25 realizations of the noisy input.)": This is confusing. 25 times 25 equals 50?

-- "Inspired by the stochastic nature of the single neuronal unit": As far as I know, in reality single-neuron dynamics is quite robustly deterministic. Synapses are known to be more noisy.

-- Equations (12), (13), and (14), (15) are exactly the same as, respectively, (5), (4) and (2), (1). Perhaps such a repetition is not needed?

Language:

-- "In addition, our approach results quite natural in terms of biological plausibility.": ungrammatical

-- "combined to a deterministic generation of the spikes": combined  compared ?

-- "bringing the neuron further from the threshold and ensuring a safety gap": also called margin</t>

Reviewer #3: The authors carefully addressed all the comments of the previous review round.

Some minor issues may be considered:

1. It would be nice to mention full-FORCE approach in the introduction in more explicit manner, since that is the most closely related algorithm published so far.

2. Not all of the source code is provide, e.g. coBa model is missing

some typos:

line 205 parenthesis

line 312 "The in the case"

7. PLOS authors have the option to publish the peer review history of their article (what does this mean?). If published, this will include your full peer review and any attached files.

Reviewer #1: No

Reviewer #2: **Yes: **Lukasz Kusmierz

Reviewer #3: **Yes: **Alexander Ivanenko

---

## [Author Response · Author response to Decision Letter 1]

12 Dec 2020

We addressed the comments of reviewers and editor in the attachted "Response to Reviewers II" document

---

## [Decision Letter · Decision Letter 2]

1 Feb 2021

Target spike patterns enable efficient and biologically plausible learning for complex temporal tasks

PONE-D-20-15736R2

Dear Dr. Muratore,

We’re pleased to inform you that your manuscript has been judged scientifically suitable for publication and will be formally accepted for publication once it meets all outstanding technical requirements.

Kind regards,

Gennady Cymbalyuk, Ph.D.

Academic Editor

PLOS ONE

Additional Editor Comments (optional):

Reviewers' comments:

Reviewer's Responses to Questions

**Comments to the Author**

1. If the authors have adequately addressed your comments raised in a previous round of review and you feel that this manuscript is now acceptable for publication, you may indicate that here to bypass the “Comments to the Author” section, enter your conflict of interest statement in the “Confidential to Editor” section, and submit your "Accept" recommendation.

Reviewer #1: All comments have been addressed

Reviewer #2: All comments have been addressed

Reviewer #3: All comments have been addressed

2. Is the manuscript technically sound, and do the data support the conclusions?

Reviewer #1: Yes

Reviewer #2: Yes

Reviewer #3: Yes

3. Has the statistical analysis been performed appropriately and rigorously? 

Reviewer #1: Yes

Reviewer #2: Yes

Reviewer #3: Yes

4. Have the authors made all data underlying the findings in their manuscript fully available?

Reviewer #1: Yes

Reviewer #2: Yes

Reviewer #3: Yes

5. Is the manuscript presented in an intelligible fashion and written in standard English?

Reviewer #1: Yes

Reviewer #2: Yes

Reviewer #3: Yes

6. Review Comments to the Author

Reviewer #1: (No Response)

Reviewer #2: Please accept my apologies for the delay. In my opinion the manuscript is now ready for the publication. For some reason, in my previous report one of the comments was cut short. Below I attach it, perhaps the Authors will find it useful.

If I understood correctly, during training s_pred and v at time t are calculated assuming that at times t_past<t according="" all="" authors="" been="" between="" consistency="" do="" dynamics="" ensure="" generated="" have="" how="" of="" pattern="" spikes="" spikes.="" target="" the="" to="" v=""></t>

Reviewer #3: The authors carefully addressed all the comments given during the previous review round. I recommend accepting the manuscript.

Minor: line 428 .. reproduce effects as ... -> such as ... ?

7. PLOS authors have the option to publish the peer review history of their article (what does this mean?). If published, this will include your full peer review and any attached files.

Reviewer #1: No

Reviewer #2: **Yes: **Łukasz Kuśmierz

Reviewer #3: **Yes: **Alexander Ivanenko

---

## [Editor Report · Acceptance letter]

3 Feb 2021

PONE-D-20-15736R2 

Target spike patterns enable efficient and biologically plausible learning for complex temporal tasks  

Dear Dr. Muratore:

I'm pleased to inform you that your manuscript has been deemed suitable for publication in PLOS ONE. Congratulations! Your manuscript is now with our production department. 

Kind regards, 

on behalf of

Dr. Gennady Cymbalyuk 

Academic Editor

PLOS ONE